# L-Shaped Slot-Loaded Stepped-Impedance Microstrip Structure UWB Antenna

**DOI:** 10.3390/mi11090828

**Published:** 2020-08-31

**Authors:** Zhong Hua Ma, Yan Feng Jiang

**Affiliations:** 1College of Information Engineering, Jimei University, Xiamen 361021, China; mzhxm@jmu.edu.cn; 2College of IoT Engineering, Jiangnan University, Wuxi 214122, China

**Keywords:** ultra-wideband antenna, step impedance, microstrip slot, voltage standing wave ratio (VSWR)

## Abstract

A stepped planar microstrip structure is proposed and demonstrated as a candidate of the ultra-wideband (UWB) antenna in the paper. In the structure, two L-shaped slots are introduced into the rectangular microstrip patch to broaden the current path at both edges of the radiating patch. The impedance bandwidth of the antenna can be extended by using the stepped impedance resonator (SIR) structure at one end of the radiation patch and connecting with the feed line. The symmetrical two I-shaped slots are loaded on the SIR microstrip to improve in-band performance and further widen the operating band. The proposed new structure can have an improvement in the in-band characteristics while extending the operating bandwidth. A broadband impedance bandwidth of 2.39 GHz to 13.78 GHz at S11 < −10 dB is demonstrated based on the proposed novel structure. The reflection coefficient and radiation characteristics are characterized in the paper. The tiny antenna, with the benefit of small area 36 mm × 23 mm, shows potential applications in ultra-wideband communication systems, wireless energy harvesting systems, and other wireless systems.

## 1. Introduction

Ultra-wideband (UWB) technology is widely considered as one of the most potential technologies in the field of wireless communications since the US Federal Communications Commission (FCC) announced that the band of commercial communication applications for ultra-wideband systems is 3.1–10.6 GHz [1,2]. UWB technology is critical in radar [3,4] and communication systems [5,6,7]. In the systems, the UWB antenna is adapted for receiving the UWB signals with many mandatory requirements, including the wide bands, good radiation characteristics, small size, and flexible assembly, etc. Currently, there are several available UWB antennas, including the logarithmic periodic structure and spiral structure UWB antenna [8,9]. The principle of those kinds of UWB antennas is easily understood. In the above antennas, there are always multiple parts integrated into the system, corresponding to the different frequency components. However, the integration would induce the distortion and stretching of the synthesized waveform. To avoid the issue, the UWB antennas with broadband monopole structures and improved Vivaldi antenna are proposed in refs [10,11,12,13], such as roll monopole and cylindrical monopole [10,11,12], as well as the 3D bent structure [13]. Those antennas show wide bandwidth and good radiation characteristics. However, the three-dimensional structure cannot be assembled into the circuit board by the direct printing approach. Recently, the planar microstrip structure UWB antenna with compact size and excellent performance is proposed [14,15,16,17]. The antenna can easily be mass-produced and be assembled on the printed circuit board (PCB), with the additional benefit of low cost. The literature [14,15] has proposed a planar UWB antenna with a monopole structure. But the detailed information on the characterization of the antenna isn’t shown clearly. In ref [16], a UWB and narrow-band converted antenna is proposed, which is composed of the microstrip fold line and two band-pass filters, showing a stable gain and impedance bandwidth. In ref [17], a microstrip elliptical ring dipole antenna is reported. However, the band performance and the size should be optimized furthermore. In refs [18,19,20,21,22,23,24,25,26], the UWB antennas with slot structure are proposed, showing optimized band performance with small size. The literature [25] has proposed a circularly polarized square slot antenna, which generates circularly polarized excitation by embedding L-shaped microstrips of unequal sizes near two diagonal corners in a square slot. The frequency band with reflection coefficient less than −10 dB ranges from 2.8 GHz to 13 GHz, and the impedance bandwidth is 10.2 GHz. The literature [26] proposes an ultra-wideband antenna with a microstrip square ring slot. The antenna feeds by fork like-tuning stub with a good radiation pattern and constant gain in the operating frequency band.

A novel UWB antenna is proposed in the paper. In the antenna, there are two inverted L-shaped slots loaded on the side of the radiating microstrip patch and the other I-shaped slots loaded on the side of the stepped impedance microstrip closing to the feed line. The feed line is connected with the one terminal of the stepped microstrip. The other side of the dielectric layer under the feed line is providing with a ground plane spaced from the radiation patch. The characterization results demonstrate that the novel structure with the reduced area improves the in-band performance of the ultra-wideband antenna and expands the impedance bandwidth.

The organization of this paper is as follows. Firstly, the various structures of UWB antennas are briefly introduced. Furthermore, the UWB antennas with the slot structure are discussed, in which mainly the size reduction and the frequency band expansion are focused on. Then, the main structure of the proposed UWB antenna in the paper is shown. The L-shaped and I-shaped slots are loaded in a radiation patch to expand the bandwidth. The influences of the structure parameters of the antenna on the performance are discussed. The fabrication of the UWB antenna on the RO 4350 is introduced in the following part, with the key results shown, including the voltage standing wave ratio (VSWR), radiation pattern, and gain. The conclusion is summarized in the last part, showing possible application areas of the proposed antenna.

## 2. Antenna Structure

The antenna structure designed in this paper is shown in Figure 1. The dielectric substrate is Rogers 4350, with the relative dielectric constant 3.66, the loss tangent 0.004, and the thickness h of substrate 0.508 mm. The size of the UWB antenna is 36 mm × 23 mm, which is smaller than the recent reports on the most of ultra-wideband antenna in Table 1 [14,15,16,17,18,19,23,24,25,26]. The feed line is connected with the side of the stepped rectangular microstrip. The radiation microstrip patch near the feed line is loaded by the two L-shaped slot structures on both sides, respectively. A stepped impedance microstrip structure is implemented between the feed line and the radiate microstrip patch, which can improve the impedance characteristics of the antenna. The microstrip patch of the stepped impedance microstrip loaded with axisymmetric I-shaped slots can improve the bandwidth and the performance in-band. A ground plate exists on the back of the dielectric layer, where the feed line is located on the top. A narrow gap is formed between the radiating patch and the feed line.

The stepped impedance resonator (SIR) structure has been reported in the design of bandpass filters, duplexers, and metamaterial unit-cell [27,28,29,30]. The SIR is a cascade of high impedance and low impedance transmission lines. The high impedance line acts as a series inductor, and the low impedance line acts as a parallel capacitor. The SIR resonator controls the harmonic frequency corresponding to the frequency response by adjusting the impedance ratio and the length of the resonator to compensate for the unequal phase velocity of the odd mode and even mode. A wider resonance frequency band can be obtained based on the structure. In Figure 2, both terminals are at the open state and are formed by two microstrip transmission lines. The two microstrip transmission lines have different characteristic impedances (*Z*_1_ and *Z*_2_) with the electrical length *θ*_1_ and *θ*_2_, respectively. The two microstrip transmission lines have different characteristic impedances and have different electrical lengths, respectively. If the effect of step discontinuity is ignored, the basic resonance condition of SIR can be derived from the input impedance of the opened terminal, which can be written as [31]:
(1)$$Z_{\mathrm{in}}=jZ_{2}\frac{Z_{2}\mathrm{tg}\theta_{2}+Z_{1}\mathrm{tg}\theta_{1}}{Z_{2}-Z_{1}\mathrm{tg}\theta_{1}\mathrm{tg}\theta_{2}}$$

According to the resonant conditions, $Z_{\mathrm{in}}=\infty$. The following formula is established.
(2)$$Z_{2}-Z_{1}\mathrm{tg}\theta_{1}\mathrm{tg}\theta_{2}=0$$

The impedance ratio is defined as:(3)$$\frac{Z_{2}}{Z_{1}}=k=\mathrm{tg}\theta_{1}\mathrm{tg}\theta_{2}$$

k is the impedance ratio.

It can be seen from Figure 2 that the total electrical length is $\theta_{T}$. The following relationship can be obtained from Equation (2):(4)$$\begin{array}{ll} \mathrm{tg}\theta_{T} & =(\mathrm{tg}\theta_{1}+\theta_{2})=\frac{\mathrm{tg}\theta_{1}+\mathrm{tg}\theta_{2}}{1-\mathrm{tg}\theta_{1}\mathrm{tg}\theta_{2}}\\ & =\frac{1}{1-k}(\mathrm{tg}\theta_{1}+\mathrm{tg}\theta_{2})\\ & =\frac{1}{1-k}(\mathrm{tg}\theta_{1}+\frac{k}{\mathrm{tg}\theta_{1}}) \end{array}$$

When *k* = 1, it is a uniform impedance transmission line resonator.

To simplify, choose $\theta_{1}=\theta_{2}=\theta_{0}$. The resonance condition of the fundamental frequency is given by
(5)$$\theta_{0}=\mathrm{arctg}\sqrt{k}$$

The spurious frequency is $f_{s}$, and the corresponding θ is $\theta_{s}$. It is obtained from $Z_{\mathrm{in}}=\infty$ and Equation (1)
(6)$$\mathrm{tg}\theta_{s}=\infty$$

The first spurious resonance frequency $f_{s}$ is given by [31]
(7)$$\frac{f_{s}}{f_{0}}=\frac{\theta_{s}}{\theta_{0}}$$

Then
(8)$$f_{s}=\frac{\pi }{2\mathrm{arctg}\sqrt{k}}f_{0}$$

In this paper, the SIR microstrip structure is used as the transition section between the feed line and the radiating patch, which is equivalent to adding a matching network to broaden the operating frequency band. Adjusting the microstrip width of the SIR is equivalent to the adjustment of the impedance ratio. A wideband design is realized by appropriately selecting the physical structure parameters.

## 3. Simulation Design Analysis

The structure shown in Figure 1 is designed and simulated by using Ansoft’s High-Frequency Structure Simulator (HFSS) software. First of all, the rectangular patch antenna is designed on a 36 mm × 23 mm Rogers 4350 substrate. The width of the feed line is 1.07 mm (*w*). Figure 3 is the simulated antenna surface current magnitude distribution. It can be seen that the intensity of the current amplitude along the edge of the slot is significantly enhanced. The reflection coefficient curve of the simulation is shown in Figure 4. The impedance bandwidth with the reflection coefficient of less than −10 dB is very narrow. The operating frequency band is from 2.03 GHz to 4.75 GHz. The bandwidth is only 2.62 GHz, and the minimum reflection coefficient is only −18 dB in the band.

In order to expand the bandwidth and to make a good transition between the radiation patch and the feed line, the SIR microstrip structure is added between the radiating patch and the feed line. Figure 5 shows the simulation curve of the antenna reflection coefficient with SIR structure. The operating frequency band is from 2.4 GHz to 11.29 GHz, with a reflection coefficient of less than −10 dB. The bandwidth is 8.89 GHz. Two zero points appear in the passband. The bandwidth is increased, and the characteristics are improved in the band.

To further broaden the operating band and improve the characteristics in-band, two symmetric L-shaped slots are implemented on the side of the rectangular patch. The simulation curve of the reflection coefficient is shown in Figure 6. The frequency is from 2.78 GHz to 13.16 GHz, while the reflection coefficient is less than −10 dB. The bandwidth is broadened by 10.38 GHz so far.

Continuously, two symmetric I-shaped slots are implemented on the side of the SIR microstrip near the feed line. Figure 7 shows the simulation results of the reflection coefficient. It can be seen that the frequency band from 2.64 GHz to 13.93 GHz shows a small reflection coefficient of less than −10 dB. In this manner, the bandwidth is broadened by 11.29 GHz, with obvious improvements on the characteristics.

The effects of different structural parameters with L-shaped and I-shaped slots are simulated in terms of the reflection coefficient. The simulation results of the reflection coefficient with varied structures are shown in Figure 8, with the changing of the length *H*_2_. It can be seen that *H*_2_ is insensitive at low-frequency, while very sensitive at high-frequency. When *H*_2_ increases from 1 mm to 3 mm, the 10 dB impedance bandwidth decreases.

When the length *L*_2_ of the other side of the L-shaped slot is varied from 0.5 mm to 1.5 mm, the simulation results of the reflection coefficients are as shown in Figure 9. With a decrease in *L*_2_, the impedance bandwidth is broadened. But the in-band characteristics are degraded with an increase in *L*_2_. At the same time, the impedance bandwidth is narrowed, and the in-band characteristics are improved.

Figure 10 shows the simulation results of the reflection coefficient curve of the antenna when the widths *w*_slot_ of the L-shaped slot are 0.1, 0.5, and 0.7 mm, respectively. As the width of the L-shaped slot is increased, the bandwidth is narrowed, and the in-band characteristics are improved. When the slot width is decreased, the bandwidth is increased accordingly. Meanwhile, the in-band characteristic is degraded.

When the distance of the I-shaped slot from the center of the feed line is fixed to be 3.91 mm, the length of the I-shaped slot is varied. The corresponding simulation results of the reflection coefficient are shown in Figure 11. With the increase in the length (*I*_slot_), the bandwidth decreases. Meanwhile, the characteristics deteriorate in the operating band. In this manner, the length of the I-shaped slot should be smaller than the height (*H*_4_) of the SIR microstrip to avoid the obvious degradation of the bandwidth.

The structural parameter S in Figure 1 is the distance between the radiation patch on the top layer of the substrate and the ground plate on the back of the substrate, which is mainly for the good broadband matching performance. Figure 12 shows the reflection coefficient results when S changes from 0.3 mm to 1.1 mm in the step of 0.2 mm. When the value of S is less than 0.7 mm, the bandwidth decreases, and the in-band characteristic deteriorates sharply. When the value of S is greater than 0.7 mm, the bandwidth is slightly reduced, but the in-band characteristics may become unsatisfactory. Therefore, when the value of S is equal to 0.7 mm, the performance of the antenna is the best.

According to the above analysis, the key geometric values of the optimized antenna are shown in Table 1. The novel UWB antenna in the paper is designed with the following geometric parameters. The two L-shaped slots are implemented on both sides of the rectangular microstrip patch, respectively, with the length of the slot 4 mm and the width 0.5 mm. The L-shaped slot is 0.5 mm from the edge of the rectangular microstrip patch. The L-shaped slot can be used to extend the path of the surface current while miniaturizing the antenna size and improving the characteristics of the antenna in the ultra-wideband.

## 4. Results

The antenna is fabricated and measured after optimizing the structure. Agilent’s vector network analyzer (VNA) (model: Agilent E8362B, Keysight, Santa Rosa, CA, USA) is used to measure in the anechoic chamber. Figure 13 shows the simulated and experimental measured results of the reflection coefficient based on the designed antenna. The frequency band with the reflection coefficient smaller than −10 dB is from 2.64 GHz to 13.93 GHz based on the simulation result. Based on the experimental measurement, the band with the reflection coefficient of less than −10 dB is from 2.39 GHz to 13.78 GHz with the bandwidth of 11.39 GHz. The frequency bandwidth for VSWR < 2 of the simulation is from 2.67 GHz to 13.52 GHz, and the measured frequency band is from 2.39 GHz to 13.57 GHz, as shown in Figure 14.

Figure 15 shows the xoz plane and yoz plane radiation patterns of far-field radiation at different frequencies of 4 GHz, 7 GHz, and 11 GHz. The simulations and measurements are shown at the same time. Based on the results, it can be seen that there is no obvious distortion of the radiation pattern on the xoz plane and yoz plane at the three frequency points. The xoz plane pattern is a directional radiation pattern, and the yoz plane pattern is close to the omnidirectional radiation pattern. Figure 16 is the measured peak gain of the proposed antenna. The peak gain of the antenna is varied in the range of 4 dBi to 8.54 dBi in the whole operating frequency band. For a clear comparison, the results of this paper are also included in Table 2. The key results of the published planar microstrip antennas are summarized in Table 2. It can be seen from Table 2 that there are obvious shortcomings for the published antennas. The UWB antenna proposed in the paper shows the best properties among the published results.

## 5. Conclusions

The ultra-wideband antenna with a stepped impedance microstrip connection between the radiating patch and the feed line is proposed and implemented in the paper. In the designed antenna, two L-shaped slot structures are loaded on both sides of the radiating patch. The measured impedance bandwidth reaches 11.39 GHz. The yoz plane radiation pattern of the antenna is close to the omnidirectional radiation mode. The peak gain of the UWB antenna is 4 dBi to 8.54 dBi in the range of 2.39 GHz to 13.78 GHz. This UWB antenna can be used in wireless sensor network nodes, radio wave energy harvesting, and portable wireless devices.

## Figures and Tables

**Figure 1 micromachines-11-00828-f001:**
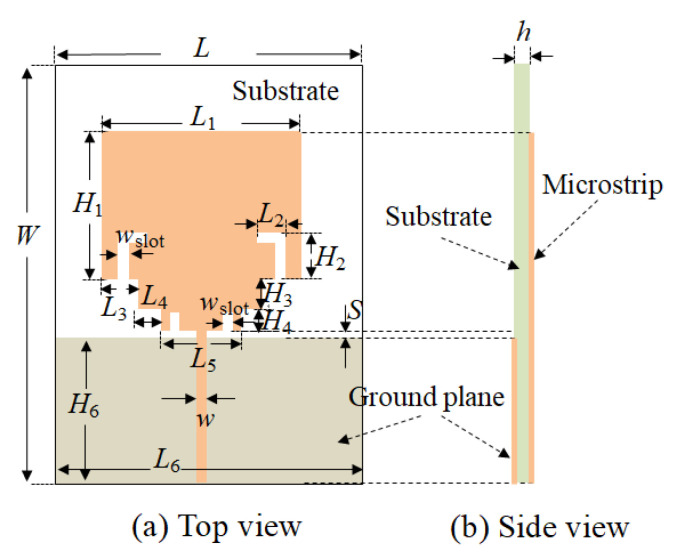
Structure of the proposed ultra-wideband (UWB) antenna. (**a**) Top view; (**b**) Side view.

**Figure 2 micromachines-11-00828-f002:**
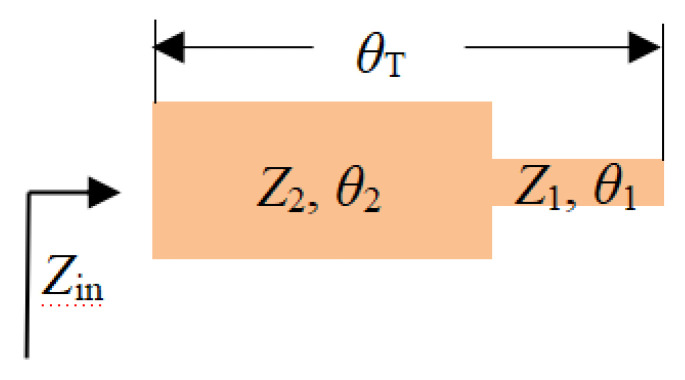
Structure of the stepped impedance resonator (SIR).

**Figure 3 micromachines-11-00828-f003:**
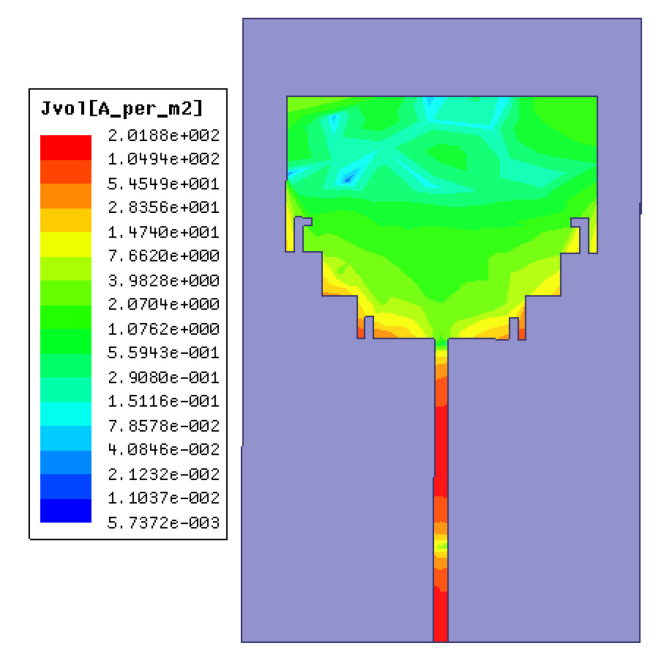
The current distribution of the antenna.

**Figure 4 micromachines-11-00828-f004:**
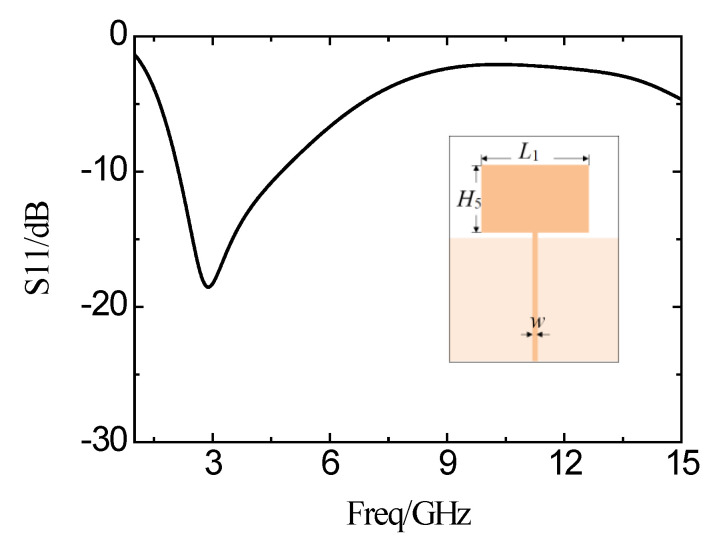
Simulation results of the reflection coefficient of the rectangular microstrip patch antenna.

**Figure 5 micromachines-11-00828-f005:**
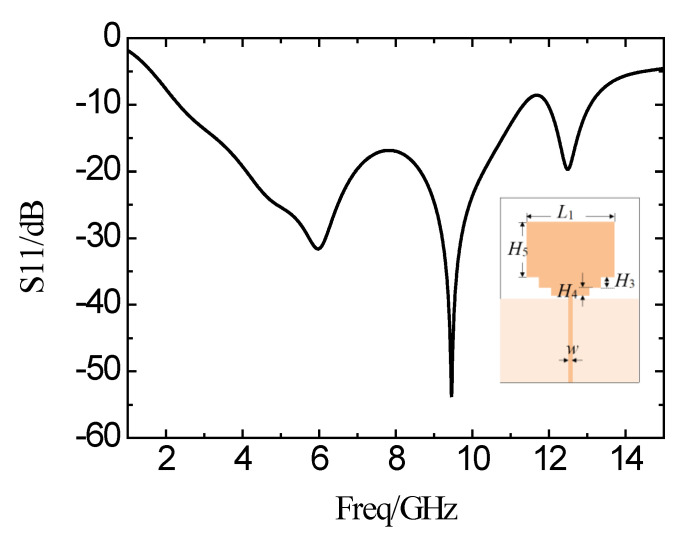
Simulation results of the reflection coefficient curve of the antenna after adding SIR structure.

**Figure 6 micromachines-11-00828-f006:**
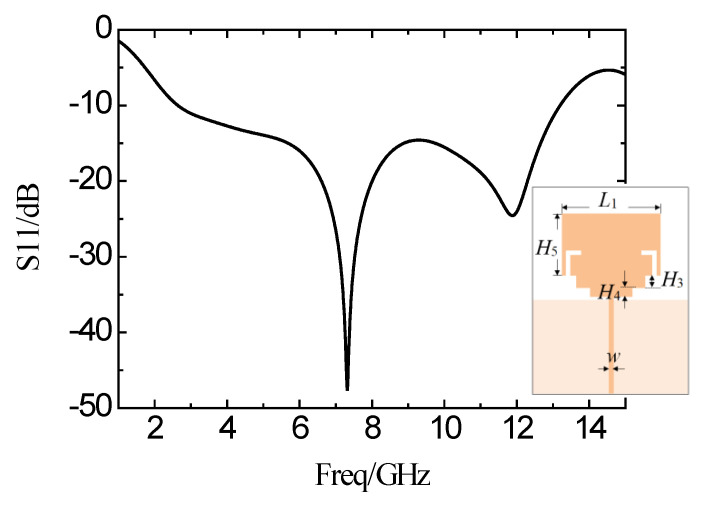
Simulation results of the reflection coefficient curve after loading two L-shaped slots on the microstrip patch.

**Figure 7 micromachines-11-00828-f007:**
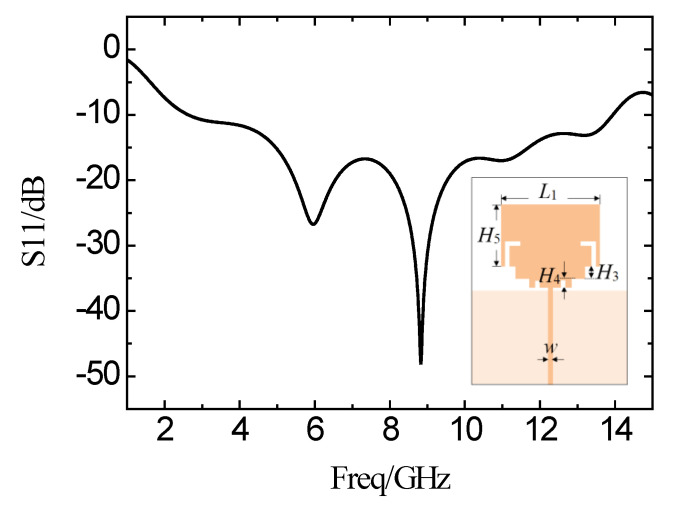
Simulation results of the reflection coefficient curve after loading the I-shaped slot on the SIR.

**Figure 8 micromachines-11-00828-f008:**
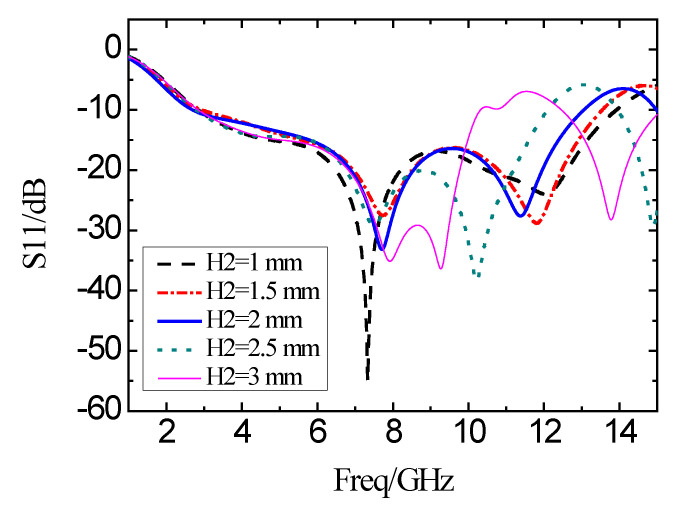
Simulation results of the reflection coefficient with different H_2_.

**Figure 9 micromachines-11-00828-f009:**
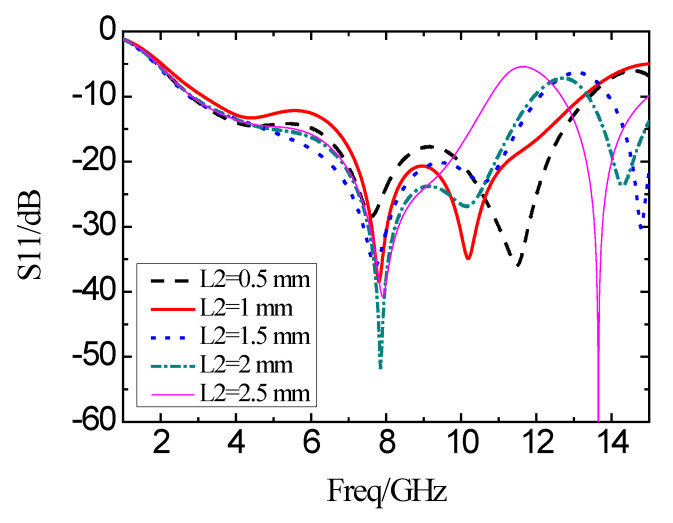
Simulation results of the reflection coefficient with different *L*_2_.

**Figure 10 micromachines-11-00828-f010:**
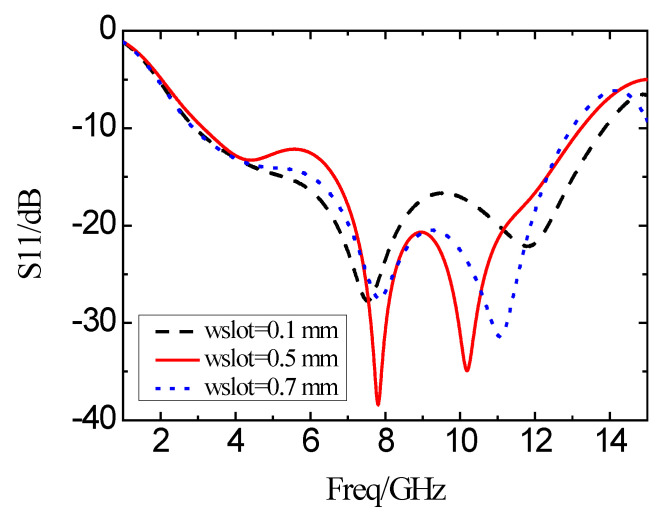
Simulation results of the reflection coefficient with different w_slot_.

**Figure 11 micromachines-11-00828-f011:**
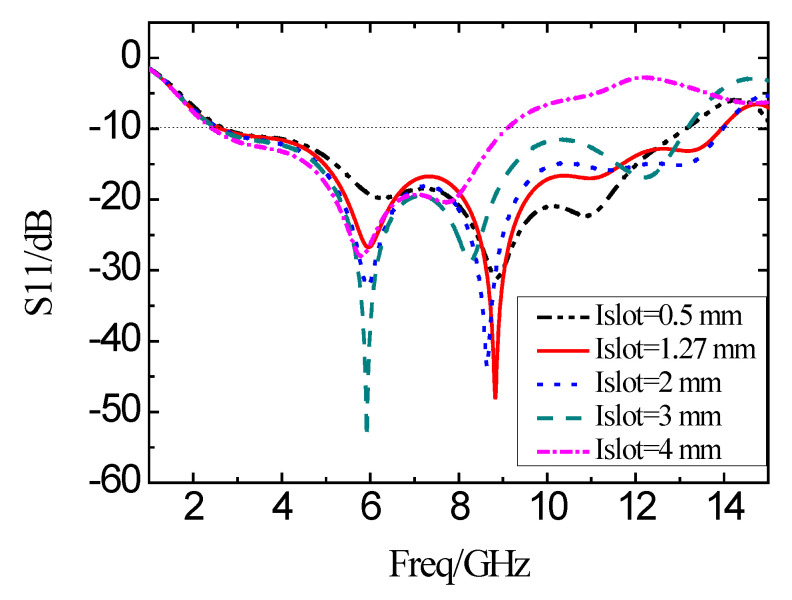
Simulation results of the reflection coefficient of antennas with the varying I-shaped slot length.

**Figure 12 micromachines-11-00828-f012:**
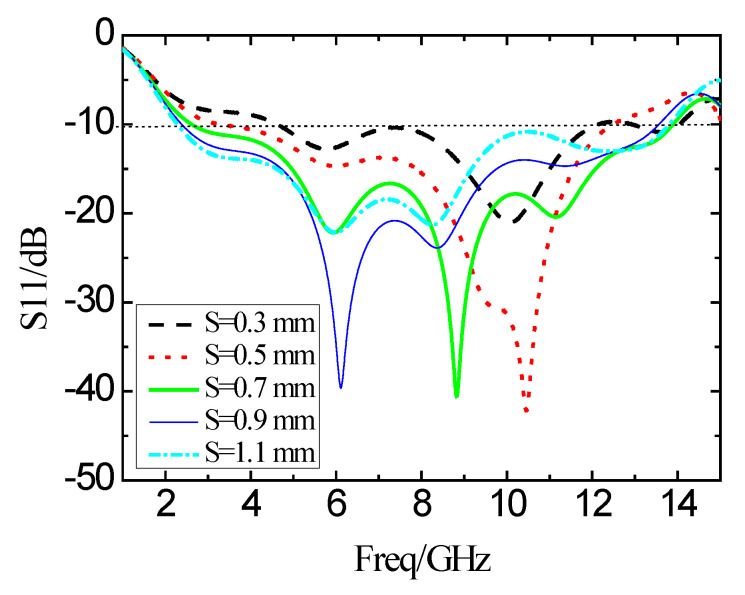
Simulation results of the reflection coefficient of the S changes.

**Figure 13 micromachines-11-00828-f013:**
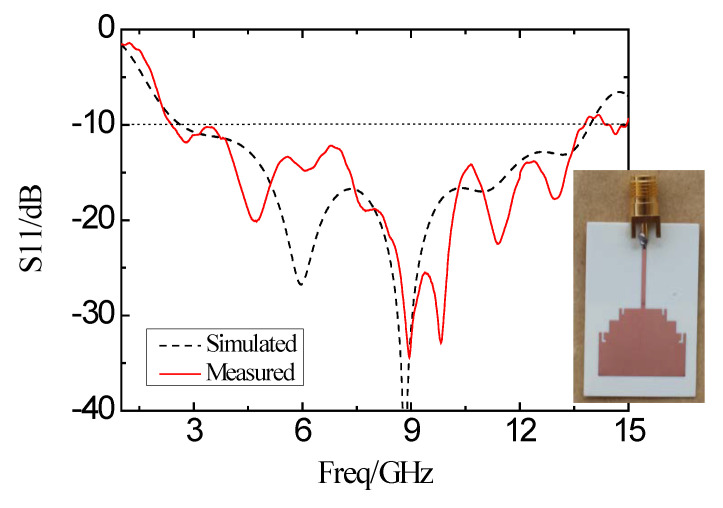
Simulation and experimental measurement of the reflection coefficient of the proposed antenna.

**Figure 14 micromachines-11-00828-f014:**
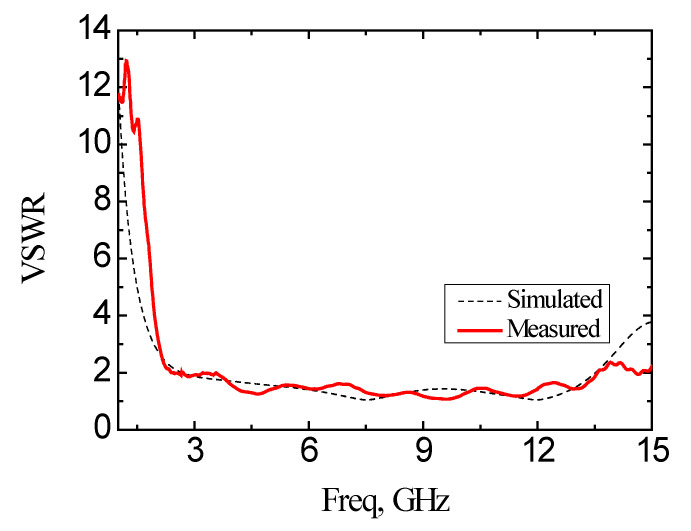
Simulation and experimental measurement of voltage standing wave ratio (VSWR) of the proposed antenna.

**Figure 15 micromachines-11-00828-f015:**
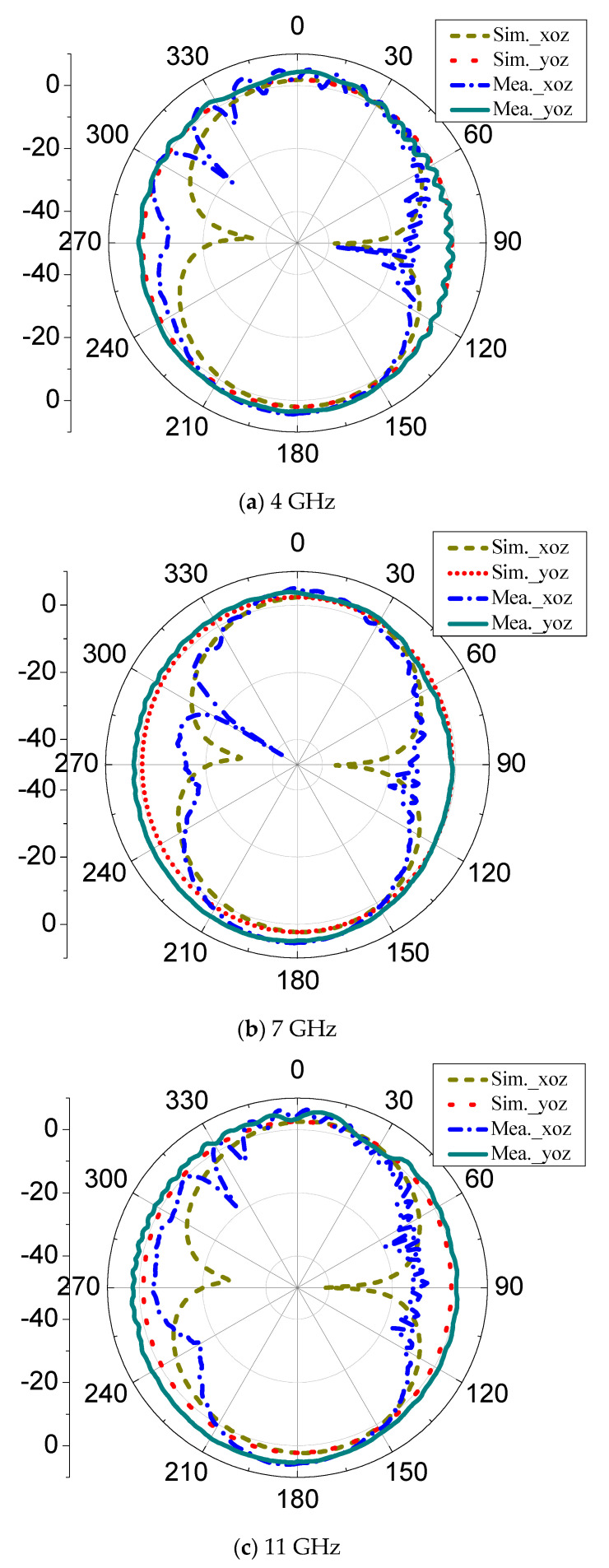
Simulation and measurement radiated pattern on xoz plane and yoz plane, respectively. (**a**) 4 GHz, (**b**) 7 GHz, (**c**) 11 GHz.

**Figure 16 micromachines-11-00828-f016:**
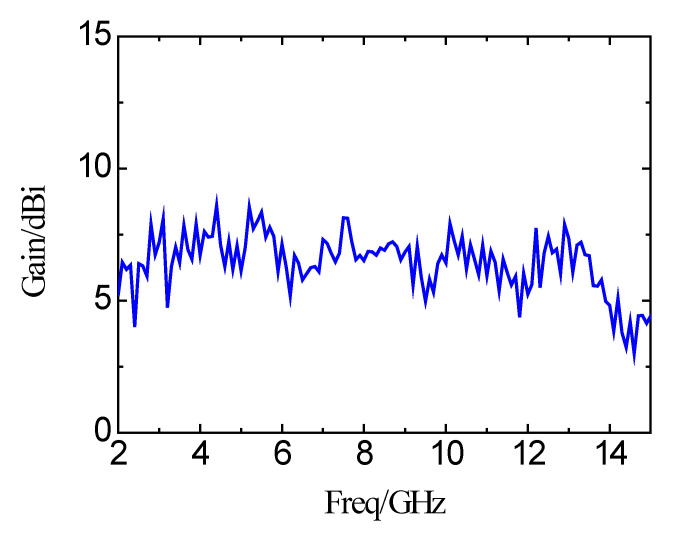
Measured peak gains of the antenna.

**Table 1 micromachines-11-00828-t001:** Parameter values of the ultra-wideband (UWB) antenna.

Parameter	Value (mm)	Parameter	Value (mm)
*L* _1_	18	*H* _1_	11
*L* _2_	1	*H* _2_	2
*L* _3_	1.465	*H* _3_	2.5
*L* _4_	4	*H* _4_	2.5
*L* _5_	11.07	*w* _slot_	0.5
*L* _6_	23	*H* _6_	16.8
*w*	1.07	*h*	0.508
*I* _slot_	1.27	*S*	0.7
*W*	36	*L*	23

*I*_slot_ is the length of the I-shaped slot.

**Table 2 micromachines-11-00828-t002:** Comparison of the published UWB antennas.

References	Type	Dielectric Constant	Size (mm^2^)	Bandwidth (GHz)	Peak Gain (dBi)	Gain Variation (dB)
[14]	Disc monopole	4.7	42 × 50	2.78–9.78	-	-
[15]	Quasi-circular	2.65	40 × 30	2.55–18.5	-	-
[16]	Microstrip	3.38	45 × 50	3–11.57	4.2–6.91	2.71
[17]	Dipole	2.2	106 × 85	1.1–11	±5	10
[18]	Reconfigurable Slot-ring array	2.2	80 × 80	1.8–3.7 and 4.5–8.23	2.4–3.1	0.7
[19]	Annular slot	3.48	120 × 60	0.75–7.65	0.5–3.2	2.7
[20]	Slot	4.4	25 × 30	2.9–11.8	2–6	4
[21]	Slot	4.4	30 × 30	4–10	0–4.8	4.8
[22]	Tapered slot	4.4	30 × 13.5	2.95–14	1.7–4.1	2.4
[23]	Slot	2.33	52 × 62	2.1–11.5	3.5–7	3.5
[24]	Square slot	2.2	50 × 50	2.1–11.5	3.8–5.7	1.9
[25]	Square slot	4.4	60 × 60	2.8–13	3–4.13	1.13
[26]	Square-ring slot	3.4	120 × 100	3–11	2–6	2
This paper	Slot	3.66	36 × 23	2.39–13.78	4–8.54	4.54

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
