# Peer review of "L-Shaped Slot-Loaded Stepped-Impedance Microstrip Structure UWB Antenna"

_micromachines, 2020, doi:10.3390/mi11090828_

Round 1
Reviewer 1 Report
1. The manuscript is not well-organized. For example, Table 1 shows a comparison between the current work and that from other sources. The table should be moved to the section of the manuscript right before the last or conclusion section.
2. The quality of table body in Table 1 needs be improved. The “slot” should be “Slot”.
3. The quality of column title in Table 2 needs be improved. It seems that the “meter” is the unit of “Para” when the “Parameter” is divided into two words and is separated in different rows.
4. Line 133 and 145: “All figures and tables should be cited in the main text as Figure 1, Table 1, etc.” The above-mentioned sentence could not belong to any part of the manuscript. The authors did not pay attention to what they were writing or copying.
5. The figure legends in the 3 subfigures in Figure 12 are too small in size to read.
6. There are three sections sharing the same section #, 3 (Line 74, 135, and 157),
7. No detailed theoretical analysis is presented for the proposed antenna in the manuscript. Once a structure involving step-impedance resonator (SIR) is specified, the corresponding resonant modes should be predicted or calculated.

Author Response
We addressed all the comments as detailed below and have updated the manuscript accordingly.
1.The manuscript is not well-organized. For example, Table 1 shows a comparison between the current work and that from other sources. The table should be moved to the section of the manuscript right before the last or conclusion section.
Answer: With respect to the reviewer’s comment- This is a very important suggestion. The Table is moved to the section before conclusion section. Table 1 is modified to be Table 2 in the revision.
(In page 11, line 215)
For a clear comparison, the results of this paper are also included in Table.2. The key results of the published planar microstrip antennas are summarized in Table 2. It can be seen from the Table.2 that there are obvious shortcomings for the published antennas. The UWB antenna proposed in the paper shows the best properties among the published results. (In page 11, line 220) Table 2. Comparison of the published UWB antennas References Type Dielectric constant Size (mm2) Bandwidth (GHz) Peak Gain (dBi) Gain Variation (dB) [7] Disc monopole 4.7 42×50 2.78-9.78 - - [8] Quasi-circular 2.65 40×30 2.55-18.5 - - [9] Microstrip 3.38 45×50 3-11.57 4.2-6.91 2.71 [10] Dipole 2.2 106×85 1.1-11 ±5 10 [11] Reconfigurable Slot-ring array 2.2 80×80 1.8-3.7 and 4.5-8.23 2.4-3.1 0.7 [12] Annular slot 3.48 120×60 0.75-7.65 0.5-3.2 2.7 [13] Slot 4.4 25×30 2.9-11.8 2-6 4 [14] Slot 4.4 30×30 4-10 0-4.8 4.8 [15] Tapered slot 4.4 30×13.5 2.95-14 1.7-4.1 2.4 [16] Slot 2.33 52×62 2.1-11.5 3.5-7 3.5 [17] Square slot 2.2 50×50 2.1-11.5 3.8-5.7 1.9 [18] Square slot 4.4 60×60 2.8-13 3-4.13 1.13 [19] Square-ring slot 3.4 120×100 3-11 2-6 2 This paper Slot 3.66 36×23 2.39-13.78 4-8.54 4.54
2.The quality of table body in Table 1 needs be improved. The “slot” should be “Slot”.
Answer: With respect to the reviewer’s comment- the response is that the word “slot” has been modified to “Slot” in the Table 2. Sorry for the ignorance.
3.The quality of column title in Table 2 needs be improved. It seems that the “meter” is the unit of “Para” when the “Parameter” is divided into two words and is separated in different rows.
Answer: With respect to the reviewer’s comment- the response is that the quality of column title in Table 1 has been improved. Sorry for the ignorance. Table 2 is modified to be Table 1 in the revised version.
(In page 8, line 180)
Table 1 Parameter values of the UWB antenna Parameter Value (mm) Parameter Value (mm) L1 18 H1 11 L2 1 H2 2 L3 1.465 H3 2.5 L4 4 H4 2.5 L5 11.07 wslot 0.5 L6 23 H6 16.8 w 1.07 h 0.508 Islot 1.27 S 0.7 W 36 L 23
4.Line 133 and 145: “All figures and tables should be cited in the main text as Figure 1, Table 1, etc.” The above-mentioned sentence could not belong to any part of the manuscript. The authors did not pay attention to what they were writing or copying.
Answer: With respect to the reviewer’s comment- Really sorry for the mistake. The two sentences are deleted in manuscript. Thank you very much for your suggestion. Delete the sentence “All figures and tables should be cited in the main text as Figure 1, Table 1, etc.” in line 133 and 145
5.The figure legends in the 3 subfigures in Figure 12 are too small in size to read.
Answer: With respect to the reviewer’s comment- Thank you for pointing it out. Authors have revised the Figure 15 to avoid reading confusion. The Figure 15 is modified in Page 10 .
(a)4GHz (b)7GHz (c)11GHz
Figure 15. Simulated and measured radiated pattern on xoz plane and yoz plane respectively. (a) 4 GHz (b) 7 GHz (c) 11 GHz.
6.There are three sections sharing the same section #, 3 (Line 74, 135, and 157),
Answer: With respect to the reviewer’s comment-Sorry for the mistake. The mistake is corrected in the revised manuscript. Thank you for pointing it out.
(In page 9, line 196)
4. Results
(In page 12, line 222)
5. Conclusion
7. No detailed theoretical analysis is presented for the proposed antenna in the manuscript. Once a structure involving step-impedance resonator (SIR) is specified, the corresponding resonant modes should be predicted or calculated
Answer: With respect to the reviewer’s comment-Thank you for pointing it out. The related content is added to the text.
(In page 3, line 86)
Figure 2. Structure of the SIR.
(In page 3, line 87)
The stepped impedance resonator (SIR) structure has been reported in the design of bandpass filters, duplexers and metamaterial unit-cell [20-22]. The SIR is a cascade of high impedance and low impedance transmission lines. The high impedance line acts as a series inductor, and the low impedance line acts as a parallel capacitor. The SIR resonator controls the harmonic frequency corresponding to the frequency response by adjusting the impedance ratio and the length of the resonator to compensate for the unequal phase velocity of the odd mode and even mode. A wider resonance frequency band can be obtained based on the structure. In Fig.2, both terminals are at open state, which are formed by two microstrip transmission lines. The two microstrip transmission lines have different characteristic impedances (Z1 and Z2 ) with the electrical length θ1 and θ2, respectively. The two microstrip transmission lines have different characteristic impedances and have different electrical lengths respectively. If the effect of step discontinuity is ignored, the basic resonance condition of SIR can be derived from the input impedance of the opened terminal, which can be written as [23]: (1) According to the resonant conditions, , The following formula is established. (2) The impedance ratio is defined as: (3) is the impedance ratio. It can be seen from Figure 2 that the total electrical length is. The following relationship can be obtained from Eqn. 2: (4) When k=1, it is a uniform impedance transmission line resonator. To simplify, choose . The resonance condition of the fundamental frequency is given by (5) The spurious frequency is and corresponding is . It obtain from and Eqn. 1 (6) The first spurious resonance frequencyis given by [23] (7) then (8) In this paper, SIR microstrip structure is used as the transition section between the feed line and the radiating patch, which is equivalent to adding a matching network to broaden the operating frequency band. Adjusting the microstrip width of the SIR is equivalent to the adjustment of the impedance ratio. A wideband design is realized by appropriately selecting the physical structure parameters.
(In page 13, line 282)
20.Hsu C.; Kuo J. Design of Cross-Coupled Quarter-Wave SIR Filters with Plural Transmission Zeros. 2006 IEEE MTT-S International Microwave Symposium Digest, San Francisco, CA, 2006, 1205-1208.
21.Lerdwanittip R.; Namsang A.; Akkaraekthalin P. Bandpass Filters Using T-shape Stepped Impedance Resonators for Wide Harmonics Suppression and Their Application for a Diplexer. Journal of Semiconductor Technology and Science, 2011, 11, 65-72.
22.Zarghooni B.; Denidni T. A. New Compact Metamaterial Unit-Cell Using SIR Technique. IEEE Microwave and Wireless Components Letters, 2014, 24,315-317.
23.Makimoto M.; Yamashita S. Bandpass Filters Using Parallel Coupled Stripline Stepped Impedance Resonators. IEEE Transactions on Microwave Theory and Techniques, 1980, 28, 1413-1417.

Reviewer 2 Report
Dear authors,
I would like to congratulate you for your interesting work entitled "L-shaped slot-loaded stepped-impedance microstrip structure UWB antenna".
Introduction section must include how the document is structured section by section after Table 1, end of section 1.
Table 1, last colum (Variation) should include some values. Maybe all antennas included in Table 1 should include the input impedance used, e.g 50 or 75 Ohms.
Parameter S in Figure 1 is not defined, neither its value included in the text. Which is the dimension of the narrow gap between the radiating patch and the feed line? How would affect the variation of this parameter in the presented results?
The ground plane of the UWB antenna proposed is located in the back side of the dielectric material. How could vary the results if this ground plane is located on the front side?
Figure 1 must include the two main dimensions of the antenna (size) although it is included in Table 1.
In section 2, there is no information, neither references, of how must be designed the stepped impedance resonator (SIR) microstrip structure. Although the results are promissing, authors must clearly explain how it is justified the proposed antenna. It is not enough to only include the results without explananing how to design the antenna. It is a big flaw in the presented research.
Some minor spelling mistakes are included in the text:
79 "frquency" instead of frequency
Figures 5 and 6 "simulated" or simulating
133 and 145 "All figures and tables should be cited in the main text as Figure 1, Table 1, etc." must be quitted.
Best regards
Author Response
We addressed all the comments as detailed below and have updated the manuscript accordingly.
I would like to congratulate you for your interesting work entitled "L-shaped slot-loaded stepped-impedance microstrip structure UWB antenna".
- Introduction section must include how the document is structured section by section after Table 1, end of section 1.
Answer: With respect to the reviewer’s comment-Thank you for pointing it out. The following paragraph is added in Page 2, line 54 to line 62
(In page 2, line 63)
The organization of this paper is as following. Firstly, the various structures of UWB antennas are briefly introduced. Furthermore, the UWB antennas with the slot structure are discussed, in which mainly focused on the size reduction and the frequency band expansion. Then the main structure of the proposed UWB antenna in the paper is shown. The L-shaped and I-shaped slots are loaded in radiation patch to expand the bandwidth. The influences of the structure parameters of the antenna on the performance are discussed. The fabrication of the UWB antenna on the RO 4350 is introduced in the following part, with the key results are shown, including the VSWR, radiation pattern and gain. The conclusion is summarized in the last part, showing possible application areas of the proposed antenna.
2.Table 1, last colum (Variation) should include some values. Maybe all antennas included in Table 1 should include the input impedance used, e.g 50 or 75 Ohms.
Answer: With respect to the reviewer’s comment-Thank you for pointing it out. The Table 1 has been modified Table 2. UWB antennas work in a wide frequency band. Furthermore, the input impedance of the antenna is a function of frequency, the impedance various in a wide frequency band. It cannot be represented by a certain value. As long as the imaginary part is small and the real part is close to the characteristic impedance, we can approximate it as matched with the characteristic impedance.
3.Parameter S in Figure 1 is not defined, neither its value included in the text. Which is the dimension of the narrow gap between the radiating patch and the feed line? How would affect the variation of this parameter in the presented results?
Answer: With respect to the reviewer’s comment-Thank you for pointing it out. The definite value of the S has been added to Table 1. The definition of the S and analysis of the S value changing have been added to the text.
(In page 8, line 181)
The structural parameter S in Fig.1 is the distance between the radiation patch on the top layer of the substrate and the ground plate on the back of the substrate, which mainly for the good broadband matching performance. Fig.12 shows the return loss results when S changes from 0.3 mm to 1.1 mm in step of 0.2 mm. When the value of S is less than 0.7 mm, the bandwidth decreases and the in-band characteristic deteriorate sharply. When the value of S is greater than 0.7 mm, the bandwidth is slightly reduced, but the in-band characteristics may become unsatisfactory. Therefore, when the value of S is equal to 0.7 mm, the performance of the antenna is the best.
(In page 8, line 176)
Figure 12. Simulation results of the return loss of the S changes.
4.The ground plane of the UWB antenna proposed is located in the back side of the dielectric material. How could vary the results if this ground plane is located on the front side?
Answer: With respect to the reviewer’s comment-Thank you for pointing it out. This is really a very interesting idea.
In our design, the ground plane is located at the back side covered with the dielectric material to ensure its UWB and radiation characteristic. If the application scenario requires the ground plane on the front side, the area of the antenna would be enlarged. Enough area of the front side would be covered with the dielectric material to keep the related characteristics of the antenna. The properties of the UWB antenna would be influenced. However, further investigation should be tried on the interesting idea before a conclusion can be made.
Thank you again for your interesting idea. We will try to investigate its feasibility in our following research work.
5.Figure 1 must include the two main dimensions of the antenna (size) although it is included in Table 1.
Answer: With respect to the reviewer’s comment-Thank you for pointing it out. The length and width of the substrate is L6 and W. The two parameters are marked in Figure 1.
(In page 2)
|
|
Figure 1. Structure of the proposed UWB antenna
6.In section 2, there is no information, neither references, of how must be designed the stepped impedance resonator (SIR) microstrip structure. Although the results are promissing, authors must clearly explain how it is justified the proposed antenna. It is not enough to only include the results without explananing how to design the antenna. It is a big flaw in the presented research.
Answer: With respect to the reviewer’s comment-Thank you for pointing it out. The related content is added to the text.
(In page 3, line 86)
|
|
Figure 2. Structure of the SIR.
(In page 3, line 87)
The stepped impedance resonator (SIR) structure has been reported in the design of bandpass filters, duplexers and metamaterial unit-cell [20-22]. The SIR is a cascade of high impedance and low impedance transmission lines. The high impedance line acts as a series inductor, and the low impedance line acts as a parallel capacitor. The SIR resonator controls the harmonic frequency corresponding to the frequency response by adjusting the impedance ratio and the length of the resonator to compensate for the unequal phase velocity of the odd mode and even mode. A wider resonance frequency band can be obtained based on the structure. In Fig.2, both terminals are at open state, which are formed by two microstrip transmission lines. The two microstrip transmission lines have different characteristic impedances (Z1 and Z2 ) with the electrical length θ1 and θ2, respectively. The two microstrip transmission lines have different characteristic impedances and have different electrical lengths respectively. If the effect of step discontinuity is ignored, the basic resonance condition of SIR can be derived from the input impedance of the opened terminal, which can be written as [23]:
(1)
According to the resonant conditions, , The following formula is established.
(2)
The impedance ratio is defined as:
(3)
is the impedance ratio.
It can be seen from Figure 2 that the total electrical length is. The following relationship can be obtained from Eqn. 2:
(4)
When k=1, it is a uniform impedance transmission line resonator.
To simplify, choose . The resonance condition of the fundamental frequency is given by
(5)
The spurious frequency is and corresponding is . It obtain from and Eqn. 1
(6)
The first spurious resonance frequencyis given by [23]
(7)
Then
(8)
In this paper, SIR microstrip structure is used as the transition section between the feed line and the radiating patch, which is equivalent to adding a matching network to broaden the operating frequency band. Adjusting the microstrip width of the SIR is equivalent to the adjustment of the impedance ratio. A wideband design is realized by appropriately selecting the physical structure parameters.
(In page 13, line 282)
- Hsu C.; Kuo J. Design of Cross-Coupled Quarter-Wave SIR Filters with Plural Transmission Zeros. 2006 IEEE MTT-S International Microwave Symposium Digest, San Francisco, CA, 2006, 1205-1208.
- Lerdwanittip R.; Namsang A.; Akkaraekthalin P. Bandpass Filters Using T-shape Stepped Impedance Resonators for Wide Harmonics Suppression and Their Application for a Diplexer. Journal of Semiconductor Technology and Science, 2011, 11, 65-72.
- Zarghooni B.; Denidni T. A. New Compact Metamaterial Unit-Cell Using SIR Technique. IEEE Microwave and Wireless Components Letters, 2014, 24,315-317.
- Makimoto M.; Yamashita S. Bandpass Filters Using Parallel Coupled Stripline Stepped Impedance Resonators. IEEE Transactions on Microwave Theory and Techniques, 1980, 28, 1413-1417.
7.Some minor spelling mistakes are included in the text:
79 "frquency" instead of frequency
Figures 5 and 6 "simulated" or simulating
133 and 145 "All figures and tables should be cited in the main text as Figure 1, Table 1, etc." must be quitted.
Answer: With respect to the reviewer’s comment-Thank you for pointing it out. The spelling mistakes are been modified in the manuscript. Really sorry for the typo.
(In page 4, line 131)
The “frquency” is modified “frequency”
(In page 6, line 154, line155)
Figure 7 and Figure 8 "simulated" is modified “Simulation results of”
(In page 7, line 139) and (In page 8, line 150)
Delete the sentence “All figures and tables should be cited in the main text as Figure 1, Table 1, etc.” in line 133 and 135
Other errors have been corrected in the text.

Reviewer 3 Report
The authors provided a detailed study of previous work on UWB antenna, but, missed several important works like the two paper shown below.
- Pourahmadazar, et al. 10.1109/LAWP.2011.2147271
- Sadat, et al. 10.1109/APS.2006.1711670
In order to support authors’ claim that L-shaped slots broadens the current path at both edges of the radiating patch, the authors are suggested to provide the simulated surface current distribution in the manuscript.
Even though the authors studied the antenna’s performance with varying I-slot length, the I-slot length of the final antenna is not mentioned. It is highly recommended to add I-slot length in table 2.
To enhance the readability, the appearance of tables and figures can be improved. For example, in table 1, variation looks like a separate column. The labels in figure 1 are too crowded. The texts in inset image of figure 2,3,4 are not readable, while the text in figure 5 are very clear.
Author Response
We addressed all the comments as detailed below and have updated the manuscript accordingly.
1.The authors provided a detailed study of previous work on UWB antenna, but, missed several important works like the two paper shown below. 10.Pourahmadazar, et al. 10.1109/LAWP.2011.2147271 11.Sadat, et al. 10.1109/APS.2006.1711670
Answer: With respect to the reviewer’s comment- This is a very important suggestion. The Table is moved to the section before conclusion section. Table 1 is modified to be Table 2 in the revision.
(In page 2, line 48)
The literature [18] proposed a circularly polarized square slot antenna, which generates circularly polarized excitation by embedding L-shaped microstrip strips of unequal sizes near two diagonal corners in a square slot. The frequency band with return loss less than -10 dB ranges from 2.8 GHz to 13 GHz, and the impedance bandwidth is 10.2 GHz. The literature [19] proposed an ultra-wideband antenna with a microstrip square ring slot. The antenna feed by fork like-tuning stub with a good radiation pattern and constant gain in the operation frequency band.
(In page 11, line 220)
Table2. Comparison of the published UWB antennas References Type Dielectric constant Size (mm2) Bandwidth (GHz) Peak Gain (dBi) Gain Variation (dB) [7] Disc monopole 4.7 42×50 2.78-9.78 - - [8] Quasi-circular 2.65 40×30 2.55-18.5 - - [9] Microstrip 3.38 45×50 3-11.57 4.2-6.91 2.71 [10] Dipole 2.2 106×85 1.1-11 ±5 10 [11] Reconfigurable Slot-ring array 2.2 80×80 1.8-3.7 and 4.5-8.23 2.4-3.1 0.7 [12] Annular slot 3.48 120×60 0.75-7.65 0.5-3.2 2.7 [13] Slot 4.4 25×30 2.9-11.8 2-6 4 [14] Slot 4.4 30×30 4-10 0-4.8 4.8 [15] Tapered slot 4.4 30×13.5 2.95-14 1.7-4.1 2.4 [16] Slot 2.33 52×62 2.1-11.5 3.5-7 3.5 [17] Square slot 2.2 50×50 2.1-11.5 3.8-5.7 1.9 [18] Square slot 4.4 60×60 2.8-13 3-4.13 1.13 [19] Square-ring slot 3.4 120×100 3-11 2-6 2 This paper Slot 3.66 36×23 2.39-13.78 4-8.54 4.54
(In page 13, line 276)
[18] Pourahmadazar J.; Ghobadi C.; Nourinia J.; et al.. Broadband CPW-Fed Circularly Polarized Square Slot Antenna With Inverted-L Strips for UWB Applications. IEEE Antennas and Wireless Propagation Letters, 2011, 10, 369-372. [19] Sadat S.; Fardis M.; Geran F.; et al.. A Compact Microstrip Square-Ring Slot Antenna for UWB Applications. 2006 IEEE Antennas and Propagation Society International Symposium, IEEE, Albuquerque, NM, USA, 2006, 4629-4632.
2.In order to support authors’ claim that L-shaped slots broadens the current path at both edges of the radiating patch, the authors are suggested to provide the simulated surface current distribution in the manuscript.
Answer: With respect to the reviewer’s comment-Thank you for pointing it out. The related content is added to the text.
(In page 4, line 127)
Figure 3 is the simulated antenna surface current magnitude distribution. It can be seen that the intensity of the current amplitude along the edge of the slot is significantly enhanced. Figure
3. Current distribution of antenna. 3.Even though the authors studied the antenna’s performance with varying I-slot length, the I-slot length of the final antenna is not mentioned. It is highly recommended to add I-slot length in table 2.
Answer: With respect to the reviewer’s comment- the response is that the quality of column title in Table 1 has been improved. Sorry for the ignorance. Table 2 is modified to be Table 1 in the revised version. Figure 11 in the text shows the results of the return loss of antennas with I-shaped slot length varying. The length of the I-shaped slot is also added to Table 1.
(In page 8, line 180)
Table 1 Parameter values of the UWB antenna Parameter Value (mm) Parameter Value (mm) L1 18 H1 11 L2 1 H2 2 L3 1.465 H3 2.5 L4 4 H4 2.5 L5 11.07 wslot 0.5 L6 23 H6 16.8 w 1.07 h 0.508 Islot 1.27 S 0.7 W 36 L 23
4.To enhance the readability, the appearance of tables and figures can be improved. For example, in table 1, variation looks like a separate column. The labels in figure 1 are too crowded. The texts in inset image of figure 2,3,4 are not readable, while the text in figure 5 are very clear.
Answer: With respect to the reviewer’s comment- the response is that the quality of column title in Table 1 has been improved. Sorry for the ignorance. Table 2 is modified to be Table 1 in the revised version. The inset images of origin Figure 2,3,4 have been improved in the text.
(In page 11, line 220)
Table 2. Comparison of the published UWB antennas References Type Dielectric constant Size (mm2) Bandwidth (GHz) Peak Gain (dBi) Gain Variation (dB) [7] Disc monopole 4.7 42×50 2.78-9.78 - - [8] Quasi-circular 2.65 40×30 2.55-18.5 - - [9] Microstrip 3.38 45×50 3-11.57 4.2-6.91 2.71 [10] Dipole 2.2 106×85 1.1-11 ±5 10 [11] Reconfigurable Slot-ring array 2.2 80×80 1.8-3.7 and 4.5-8.23 2.4-3.1 0.7 [12] Annular slot 3.48 120×60 0.75-7.65 0.5-3.2 2.7 [13] Slot 4.4 25×30 2.9-11.8 2-6 4 [14] Slot 4.4 30×30 4-10 0-4.8 4.8 [15] Tapered slot 4.4 30×13.5 2.95-14 1.7-4.1 2.4 [16] Slot 2.33 52×62 2.1-11.5 3.5-7 3.5 [17] Square slot 2.2 50×50 2.1-11.5 3.8-5.7 1.9 [18] Square slot 4.4 60×60 2.8-13 3-4.13 1.13 [19] Square-ring slot 3.4 120×100 3-11 2-6 2 This paper Slot 3.66 36×23 2.39-13.78 4-8.54 4.54
(In page 2)
Figure 1. Structure of the proposed UWB antenna
(In page 5, line 135)
Figure 3. Simulation results of the return loss of the rectangular microstrip patch antenna.
(In page 5, line 136)
Figure 4. Simulation results of the return loss curve of antenna after adding SIR structure.
(In page 5, line 137)
Figure 5. Simulation results of the return loss curve after loading two L-shaped slot on microstrip patch.
(In page 6, line 154)
Figure 6. Simulation results of the return loss curve after loading the I-shaped slot on the SIR.

Round 2
Reviewer 1 Report
- The review comments are responded accordingly. However, typo errors still need to be corrected.
- In many cases, there should be a space between a value (or a number) and the units used.
Reviewer 2 Report
Dear authors,
Thank you for addressing my concerns.
Best regards